# *Moraxella nonliquefaciens* and *M. osloensis* Are Important *Moraxella* Species That Cause Ocular Infections

**DOI:** 10.3390/microorganisms7060163

**Published:** 2019-06-04

**Authors:** Samantha J. LaCroce, Mollie N. Wilson, John E. Romanowski, Jeffrey D. Newman, Vishal Jhanji, Robert M. Q. Shanks, Regis P. Kowalski

**Affiliations:** 1Department of Ophthalmology, Wake Forest University School of Medicine, Winston-Salem, NC 27157, USA; sjlacroce@gmail.com; 2Clinical Laboratory―Microbiology, University of Pittsburgh Medical Center, Pittsburgh, PA 16148, USA; wilsonm12@upmc.edu; 3The Charles T. Campbell Ophthalmic Microbiology Laboratory, Department of Ophthalmology, University of Pittsburgh School of Medicine, Pittsburgh, PA 15213, USA; jer157@pitt.edu (J.E.R.); jhanjiv@upmc.edu (V.J.); shanksrm@upmc.edu (R.M.Q.S.); 4Department of Biology, Lycoming College, Williamsport, PA 17701, USA; newman@lycoming.edu

**Keywords:** *Moraxella*, keratitis, conjunctivitis, endophthalmitis, eye infections, DNA sequencing, MALDI-TOF MS, Biolog

## Abstract

*Moraxella* is an ocular bacterial pathogen isolated in cases of keratitis, conjunctivitis, and endophthalmitis. Gram-negative brick-shaped diplobacilli from ocular specimens, and slow growth in culture, are early indications of *Moraxella* ocular infection; however, identifying *Moraxella* to species can be complex and inconsistent. In this study, bacteria consistent with *Moraxella* were identified to species using: (1) DNA sequencing coupled with vancomycin susceptibility, (2) MALDI-TOF mass spectrometry, and (3) the Biolog ID system. Study samples consisted of nine ATCC *Moraxella* controls, 82 isolates from keratitis, 21 isolates from conjunctivitis, and 4 isolates from endophthalmitis. The ATCC controls were correctly identified. For keratitis, 66 (80.5%) were identified as *M.*
*nonliquefaciens*, 7 (9.0%) as *M. lacunata*, 5 (6%) as *M. osloensis*, 2 (2.5%) as *Acinetobacter lwoffii*, 1 (1.0%) as *M.*
*bovis/nonliquefaciens*, and 1 (1.0%) as *M.*
*osloensis/nonliquefaciens*. For conjunctivitis, 9 (43.0%) were identified as *M.*
*osloensis*, 6 (29.0%) as *M. nonliquefaciens*, 3 (14.3%) as *Roseomonas*, 2 (9.5%) as *Acinetobacter* (*parvus, junii*), and 1 (4.5%) as *M. catarrhalis/nonliquefaciens*. From endophthalmitis, 3 of 4 of the isolates were *M.*
*nonliquefaciens*. Overall, *M*. *nonliquefaciens* and *M.*
*osloensis* were identified in 70% (75 of 107) and 13% (14 of 107) of cases, respectively, totaling 83% (89 of 107). *M. nonliquefaciens* and *M.*
*osloensis* are important bacterial pathogens of the eye as determined by DNA sequencing, MALDI-TOF MS, and Biolog. Although *Moraxella*
*catarrhalis* is a clinical pathogen, other species of *Moraxella* appear to have a prominent role in eye infections.

## 1. Introduction

As first described by Morax [1], the genus, *Moraxella*, appears to have a specific tropism as an ocular pathogen for keratitis [2,3,4,5], conjunctivitis [6], and endophthalmitis [7]. *Moraxella* keratitis appears to be on the rise based on recent reports [2,3,4,5,8]. Bacterial conjunctivitis is generally self-limiting, but *Moraxella* conjunctivitis can persist for weeks and presents as a follicular conjunctivitis which can be misdiagnosed as inclusion conjunctivitis due to *Chlamydia* [6,9,10,11]. *Moraxella* has been reported as an ocular pathogen in Japan [2,4,12], the United States [5,6,7,9,10], and Pakistan [13]. Although quite susceptible to ophthalmic topical antibiotics as highlighted on our frequently updated website (available online: http://eyemicrobiology.upmc.com/Antibiotic.htm), *Moraxella* keratitis can persist and induce severe inflammation resulting in hypopyon formation. Like most bacteria, with entrance to the inner eye, *Moraxella* can also cause endophthalmitis [7].

In the clinical laboratory, *Moraxella* is easily identified to genus by its classic appearance as a Gram-negative diplobacilli (brick shaped) which is easily distinguished from other Gram-negative bacteria (Figure 1), and this may be the only indication of *Moraxella* infections due to antibiotic pretreatment or the fastidious nature of *Moraxella*. *Moraxella* can retain crystal violet during Gram-staining and it is classified as oxidase-positive, non-glucose fermenting bacteria. Clinically, ocular *Moraxella* isolates are generally only identified to the genus as *Moraxella*, because it is difficult and complex to classify *Moraxella* to species using phenotypic testing. There are multiple species of *Moraxella*, but the most noted, *M. catarrhalis*, formerly known as *Branhamella catarrhalis*, is not the only species implicated in ocular infections.

Matrix assisted laser desorption/ionization time-of-flight mass spectrometry (MALDI-TOF MS) and molecular techniques such as DNA sequencing can identify bacteria to genus and species with more consistent results [14,15]. A commercial system, Biolog GenIII (Hayward, CA, USA), utilizing 96 substrates and controls, can also be used to identify bacteria to genus and species. The objective of the present study was to more precisely identify ocular isolates of *Moraxella* to include species level differentiation. It is important to recognize these bacteria as pathogens, but the objective of this study was not as a clinical study to correlate specific *Moraxella* species to clinical presentations. The keratitis, endophthalmitis, and conjunctivitis were treated as infections caused by *Moraxella*.

## 2. Methods 

### 2.1. Laboratory Diagnosis of Ocular Moraxella Infection

The ophthalmic microbiology laboratory at the University of Pittsburgh Medical Center, Pittsburgh, PA, is a fully certified clinical laboratory only dedicated for the diagnosis and treatment of ocular infection. (Available online: http://eyemicrobiology.upmc.com) Samples for keratitis (cornea) were collected using soft-tipped applicators, spatula, surgical blades, and jeweler’s forceps. Conjunctival samples were generally obtained with soft-tipped applicators, and samples to detect endophthalmitis were obtained by tapping the vitreous humor with a needle and syringe. The initial diagnostic test to detect infection from ocular samples were the Gram and Giemsa stains which can provide rapid and definitive identification of an etiologic pathogen. All corneal ulcers that were either central or were over 3 mm in size of infiltration were scraped for microscopic examination and isolation of pathogens. The decision to perform laboratory studies was a judgment call by the attending ophthalmologist.

*Moraxella* can be isolated in culture on trypticase soy agar supplemented with 5% sheep blood (5% SB) and chocolate agar. Our laboratory recommends that ocular collection be placed directly to the media. Growth is better on 5% SB, but growth is generally slow with colonies not appearing until 48 h of incubation at 37 °C in 6% CO_2_. The initial appearance can be pinpoint colonies with larger colonies (2–3 mm) appearing after 2 days. Most *Moraxella* (not *M*. *catarrhalis*) isolated from ocular specimens will appear as grey to white colonies, often with a clearer beach and umbonate colony morphology, giving a fried egg appearance (Figure 2). To be included in this study, the *Moraxella* isolates had to be cultured from the cornea, conjunctiva, or an intraocular tap.

### 2.2. Antibiotic Susceptibility of Ocular Moraxella

In vitro antibiotic susceptibility for ocular isolates were performed using Kirby–Bauer disk diffusion [16] on Mueller Hinton agar supplemented with 5% sheep blood. *Moraxella* susceptibility cannot be determined on regular Mueller Hinton agar because of the need for red blood cells for proper growth. It is important to note that ocular *Moraxella* infections are not treated systemically (i.e., orally or intravenously). Ocular *Moraxella* and other bacterial pathogens of keratitis are treated topically with antibiotics or intravitreally by direct injection in cases of endophthalmitis. There are no standards for susceptibility interpretation of topical or intravitreal treatment, but the Clinical and Laboratory Standards Institute (CLSI) standards can be used to guide therapy if the concentration of antibiotics in the ocular tissue is assumed to be equal or greater than in the serum [16]. A routine keratitis testing battery of antibiotics for patient topical treatment consists of bacitracin, vancomycin, gentamicin, ciprofloxacin, ofloxacin, polymyxin B, cefazolin, tobramycin, sulfacetamide, moxifloxacin, and cefoxitin. In general, *Moraxella* isolates test susceptible to these antibiotics (available online: http://eyemicrobiology.upmc.com/AntibioticSusceptibilities/Keratitis.htm). A zone greater of 18 mm denotes susceptibility to vancomycin. 

### 2.3. Moraxella Study Isolates

Since 1993, all bacterial pathogens isolated at University of Pittsburgh Medical Center (UPMC) from keratitis, conjunctivitis, and endophthalmitis have been stored at −80 °C with broth containing 15% glycerol for validation of new testing and patient treatment. These isolates constituted an Ocular Clinical Tissue Bank in which the isolates were de-identified to comply with Institutional Review Board (IRB) protection of patient identity. Clinical presentation data, patient identity and demographics were not tabulated for any of the bacterial isolates.

The collection was reviewed for the retrieval of *Moraxella* isolates from keratitis, conjunctivitis, and endophthalmitis. No patient contact was involved in this study (University of Pittsburgh, Institutional Review Board # PRO17050362). None of these isolates (except *M. catarrhalis*) had been identified to species. 

Our identifications also included nine American Type Culture Collection (ATCC) controls: *M. bovis* ATCC^T^ 10900, *M. caviae* ATCC^T^ 14659, *M. cuniculi* ATCC^T^ 14688, *M. nonliquefaciens* ATCC^T^ 19975, *M. osloensis* ATCC^T^ 19976, *M. atlantae* ATCC^T^ 29525, *M. lincolnii* ATCC^T^ 51388, *M. lacunata* ATCC^T^ 17967, and *Moraxella* (*Branhamella*) *catarrhalis* ATCC^T^ 24250.

### 2.4. DNA Sequencing

Presumptive *Moraxella* isolates were retrieved from −80 °C and streaked onto 5% SB and incubated at 37 °C in a 6% CO_2_ incubator for 48 h. The isolates were Gram-stained to presumptively identify as Gram-negative diplobacilli. If identified as Gram negative diplobacilli, the isolates were passaged onto new 5% SB and incubated at 37 °C in a 6% CO_2_ incubator for 48 h. The DNA from the isolates were extracted using Epicentre DNA Extraction solution Quick Extract (Madison, WI, USA). PCR was performed with Taq DNA polymerase (New England Biolabs, Ipswich, MA, USA) and the 16S rRNA gene sequence was amplified (95 °C for 5 min, 33 cycles of 95 °C for 30 s, 50 °C for 15 s, 72 °C for 1 min, followed by a 10 min extension at 72 °C) using primers 27F and 1492R [17]. If a single band was visualized by gel electrophoresis of the PCR products, the amplicons were purified using Qiagen QIAquick PCR Purification kit (Hilden, Germany). The PCR products were each analyzed with a single Sanger sequencing reaction using primer 330F at the University of Pittsburgh Genomics Research Core [18]. DNA sequences were initially compared to the NCBI non-redundant nucleotide database using BLASTN [17,19,20]. After initial screening, sequences were aligned with type strain sequences using ClustalW and a neighbor joining tree [21] was constructed using the Kimura 2 parameter model [22] and 1000 bootstrap replicates [23]. Due to differing lengths of sequence obtained, all positions with less than 50% site coverage were eliminated. Evolutionary analyses were conducted in MEGA7 [24].

### 2.5. MALDI-TOF MS

As with DNA sequencing, presumptive *Moraxella* isolates were retrieved from −80 °C and streaked onto 5% SB. The isolates were incubated at 37 °C in a CO_2_ incubator for 48 h. The isolates were Gram-stained to presumptively identify as Gram-negative diplobacilli, passaged to another 5% SB for 24 h, and delivered for MALDI-TOF MS testing (UPMC, Clinical Microbiology). 

Colony material from each isolate was transferred to a polished steel target (Bruker Daltonik, Bremen, Germany) using a clean toothpick. One microliter of Matrix (Bruker HCCA (α-cyano-4-hydroxycinnamic acid in 50% acetonitrile with 2.5% trifluoroacetic acid)) was applied within an hour and air dried for 10 min. The target was analyzed using the Bruker Microflex LT/SH MALDI-TOF instrument and Bruker Biotyper software version 4.1 with the MBT BDAL library containing 7854 mass spectra. The library contained 22 species of *Moraxella*. Spectra were obtained after 240 laser shots yielding spectra with mass/charge (*m*/*z*) ratios between 2k and 20k Da. Measurements meeting the quality criteria (log score ≥ 1.8) and a log score > 0.2 between different species were deemed acceptable identifications. Samples with scores below this cutoff or with <0.2 log between different species were retested using the formic acid tube extraction method as previously described [25]. A control organism (*E. coli* ATCC 25922 or *P. aeruginosa* ATCC 27853) was extracted and analyzed once each day to ensure the extraction procedure yielded successful identification.

### 2.6. Biolog Gen III Plate

The Biolog identification system was performed according to manufacturer’s protocol (Biolog, GEN III MicroPlate ™, Instructions for Use, available online: www.biolog.com). As with MALDI-TOF MS and DNA sequencing, presumptive *Moraxella* isolates were retrieved from −80 °C and streaked onto 5% SB. The streaks were passed to 5% SB and testing was performed on 24 h growth. Biolog testing was performed on a GEN III MicroPlate™ which contained 94 biochemicals consisting of 71 carbon source utilization assays, 23 chemical sensitivity assays, a negative control, and a positive control. These provided a phenotypic fingerprint for species identification by utilizing tetrazolium redox dyes to colorimetrically indicate carbon utilization or resistance to inhibitory chemicals. 

In brief, a medium for fastidious bacteria (IF-C, Biolog) was inoculated with a *Moraxella* isolate to a turbidity of 65% transmittance measured by a Turbidimeter (Biolog). The inoculum was aliquoted to the microplate using a reservoir and multipipetor to 96 wells at a volume of 0.1 mL per well. The plate was incubated at 34 °C and read manually for color changes at 4, 8 and 20 h. The tabulated data at each time point was entered into the Biolog’s Identification Systems Software (OOP 188rG Gen III Database v2.8). The database contained 14 species of *Moraxella*. Species identification was determined as the most probable as indicated by the software. It must be noted that there is an automated system for reading the plates for color changes. Costs dictated the manual approach for this study.

### 2.7. Identification of Moraxella to Species

Isolates of *M. catarrhalis* were not of prime interest in this study because these isolates can be identified to genus and species using standard laboratory methods. *M. catarrhalis* are Gram-negative diplococci (not diplobacilli) closely resembling *Neisseria* but are oxidase-positive, fast growing, form friable movable tan-like colonies, and are resistant to vancomycin [26,27]. Our laboratory had previously used the API NH (bioMérieux, La Balme-les-Grottes, France) system to accurately confirm *M. catarrhalis* [28].

In this study, all presumptive *Moraxella* isolates were observed to be Gram-negative diplobacilli. There was some variability in the size of the bacilli. Whereas most were brick-shaped, boxcar bacilli, some were thinner bacilli and diplococcobacilli. Some isolates retained crystal violet staining which is a characteristic of *Moraxella*. As noted previously, the colonies initially appeared as pinpoint colonies with larger colonies (2–3 mm) appearing after 2 days. These colonies appeared as grey to white, often with a clearer beach that give a fried egg appearance (Figure 2). Susceptibility to vancomycin indicated a *Moraxella* species other than *M. catarrhalis.*

## 3. Results

The laboratory records (1993–2017) indicated that there were nine cases of keratitis, 18 cases of conjunctivitis, and 0 cases of endophthalmitis caused by *M. catarrhalis*. These *Moraxella* isolates and ATCC controls of *M. caviae*, *M. cuniculi*, *M. catarrhalis* (all once part of *Branhamella*), and most *M. osloensis* were vancomycin resistant.

All control ATCC isolates were identified correctly by DNA sequencing coupled with vancomycin susceptibility, MALDI-TOF MS, and Biolog Gen III plates. The vancomycin zone of inhibition for *M. atlantae* was 16 mm, and zones of inhibition were not clear for *M. lincolnii* and *M. lacunata*. 

Table 1 summarizes the identification of *Moraxella* from keratitis, conjunctivitis, and endophthalmitis using DNA sequencing with vancomycin susceptibility, MALDI-TOF MS, and Biolog Gen III plates. Identification was reported for 82 cases of keratitis, 21 cases of conjunctivitis, and four cases of endophthalmitis. The identification of DNA sequencing was more closely associated with MALDI-TOF MS (106 of 116) than with the identification of DNA sequencing with Biolog (78 of 116) (*p* = 0.001, Fisher’s Exact) and MALDI-TOF MS with Biolog (87 of 116) (*p* = 0.005, Fisher’s Exact). This included the controls and ocular isolates, and all species of *Moraxella.*

Table 2 details the prevalence of *Moraxella* species for keratitis, conjunctivitis, and endophthalmitis. Identification of *Moraxella* to species was based on DNA sequencing coupled with vancomycin susceptibility and MALDI-TOF MS.

Many isolates could not be identified by 16S rRNA sequencing alone. In the segment of the 16S rRNA gene sequenced in this study, the *M. catarrhalis* and *M. nonliquefaciens* type strains are identical (Figure 3), but can be distinguished based on vancomycin resistance in *M. catarrhalis* and susceptibility in *M. nonliquefaciens*. The sequences from strains K127, K1630, K1664, K2450, K2695, and K2757 clustered together, but were essentially equidistant from *M. bovis, M. bovoculi, M. caprae, M. equi,* and *M. lacunata,* suggesting that they may be members of a new species requiring further analysis. Sequence analysis could differentiate *M. atlantae, M. caviae, M. lincolnii*, and *M. osloensis.* DNA sequencing was complemented with vancomycin susceptibility to identify most *Moraxella* species. Vancomycin resistant isolates were associated with *M. catarrhalis* and *M. osloensis*.

For keratitis, 66 (80.5%) were identified as *M. nonliquefaciens*, seven (9.0%) as *M. lacunata,* five (6.0%) as *M. osloensis*, two (2.5%) as *Acinetobacter lwoffii*, one (1.0%) as *M. bovis/nonliquefaciens*, and one (1.0%) as *M. osloensis/nonliquefaciens*. All of the *M. nonliquefaciens* were susceptible to vancomycin, while three of the *M. osloensis* were susceptible to vancomycin and two were resistant. Although *M. osloensis* can be resistant to vancomycin, our study indicates that this may not be a consistent characteristic. On the closer laboratory examination of the two *Acinetobacter* isolates, both were initially classified as *Moraxella* based on the observation of diplococcobacilli on Gram stain.

For conjunctivitis, nine (43.0%) were identified as *M. osloensis*, six (29.0%) as *M. nonliquefaciens*, three (14.0%) as *Roseomonas mucosa*, two (9.5%) as *Acinetobacter*, and one (4.5%) as *M. catarrhalis/nonliquefaciens*. Seven of the *M. osloensis* isolates were resistant to vancomycin and two were susceptible, while vancomycin resistance was noted for one *M. nonliquefaciens* isolates with five susceptible. The three *Roseomonas mucosa* isolates were re-examined and all three were Gram-negative diplobacilli consistent with *Moraxella*, but all three presented as pinkish highly mucoid colonies slightly different from colonies observed with *Moraxella*. Once again, on closer examination, both *Acinetobacter* isolates were diplococcobacilli on Gram stain.

From endophthalmitis, three of four (75%) of the isolates were *M. nonliquefaciens* with all three isolates susceptible to vancomycin. The lone *Neisseria shayeganii* isolate (identified by DNA sequencing only) from endophthalmitis was observed to be Gram-negative diplococcoid, oxidase-positive, and vancomycin resistant. 

Overall, *M. nonliquefaciens* and *M. osloensis* were identified in 70% (75 of 107) and 13% (14 of 107) of cases, respectively, totaling 83% (89 of 107) (Table 2).

## 4. Discussion

DNA sequencing coupled with vancomycin susceptibility, MALDI-TOF MS, and Biolog GenIII plates have given us more diagnostic options to identify *Moraxella* to species when combined with established laboratory tests such as Gram stain, culture isolation, and susceptibility testing. Gram stain and culture provided us with classical laboratory characteristics. Vancomycin provided us an additional test for separation and identification of *Moraxella* to species. *M. liquefaciens* was generally found to be susceptible to vancomycin and *M. osloensis* was found to be more resistant. 

Biolog using GenIII plates was more problematic in confidently identifying *Moraxella* to species based on manual interpretation. We were required to repeat testing for many isolates, because the controls were not positive or negative as expected and identification was not conclusive. It may be that the automated Biolog system without human interpretation would be more definitive for identification. We had more confidence with DNA sequencing coupled with vancomycin susceptibility and MALDI-TOF MS for identifying the *Moraxella* isolates to species.

It may be unusual to some that we are testing the susceptibility of Gram-negative bacteria to vancomycin, but vancomycin is part of a broad-spectrum battery of antibiotics used in our laboratory to “guide” the topical treatment of bacterial keratitis. In general, the topical empiric treatment of *Moraxella* keratitis is fortified tobramycin (14 mg/mL) and cefazolin (25 mg/mL). Some ophthalmologists may use topical vancomycin (25 mg/mL) instead of cefazolin or use topical monotherapy with a commercially available fluoroquinolone (moxifloxacin, ciprofloxacin). From a previous study [6], we suggested topical tobramycin (3 mg/mL) to be used as treatment for conjunctivitis, but other commercial topical antibiotics can be used (i.e., fluoroquinolones). Endophthalmitis is treated by direct intravitreal injection with vancomycin (1 mg) and amikacin (0.4 mg) or ceftazidime (2 mg). From the website, it needs to be noted that the *Moraxella* identified in this study were highly susceptible not only to vancomycin (97%–100%), but to the aminoglycosides (94%–100%) (gentamicin, tobramycin, and amikacin), fluoroquinolones (100%) (ciprofloxacin, ofloxacin, and moxifloxacin), polymyxin B (99%–100%), and cefazolin (98%). *Moraxella* was less susceptible to trimethoprim (11%). Susceptibility could not be reported for the individual *Moraxella* species due to the de-identified nature of the data (available online: http://eyemicrobiology.upmc.com).

Our study determined that *M. nonliquefaciens* is the predominant species isolated from keratitis and *M. osloensis* is a frequent species implicated in conjunctivitis. A small sample (three of four) demonstrated that *M. nonliqufaciens* was implicated in endophthalmitis, and this has been reported previously [7]. Our study also indicated that *Roseomonas mucosa* has characteristics similar to *Moraxella* and there needs to be close examination of cultures and diagnostic testing to definitively distinguish these two genera. We have designated *Roseomonas mucosa* as a conjunctivitis pathogen, and it has been previously reported to be a causative agent in keratitis [29] and endophthalmitis [30,31].

The literature has sparse reports of ocular *Moraxella* infections, but there is no definitive consistent method to precisely identify *Moraxella* ocular isolates to genus and species. It must be noted that Graham et al. [32] using non-molecular methods, reported that eye infections were mostly associated with *M. nonliquefaciens*, *M. lacunata*, *M. osloensis*, and *M. atlantae*, supports our study results. A recent report of 17 cases of keratitis accurately identified four cases of *M. catarrhalis*, one case of *M. osloensis*, and 12 cases of *Moraxella* species with the VITEK 2 system using Gram-negative cards (SYSMEX bioMѐrieux, Tokyo, Japan) with ID-test HN20 Rapid Kit (Nissui Pharmaceutical, Tokyo, Japan) [2]. Public Health England has provided an algorithm for identification of *Moraxella*, not specifically for ocular isolates, that included a battery of laboratory tests and the possible introduction of mass spectrometry and nucleic acid amplification testing [33]. Automated systems and test kits can be accurate methods to identify ocular isolates of *Moraxella* to genus for expedient patient care. Advances in technology now allow for more precise and consistent methods to correlate specific ocular clinical presentations to species of *Moraxella*.

In conclusion, our study has identified *M. liquefaciens*, *M. osloensis*, and other *Moraxella* species as ocular pathogens. DNA sequencing coupled with vancomycin susceptibility and MALDI-TOF MS are reliable methods for the identification of *Moraxella* to species, but added investigation with automation may be required to validate Biolog.

## Figures and Tables

**Figure 1 microorganisms-07-00163-f001:**
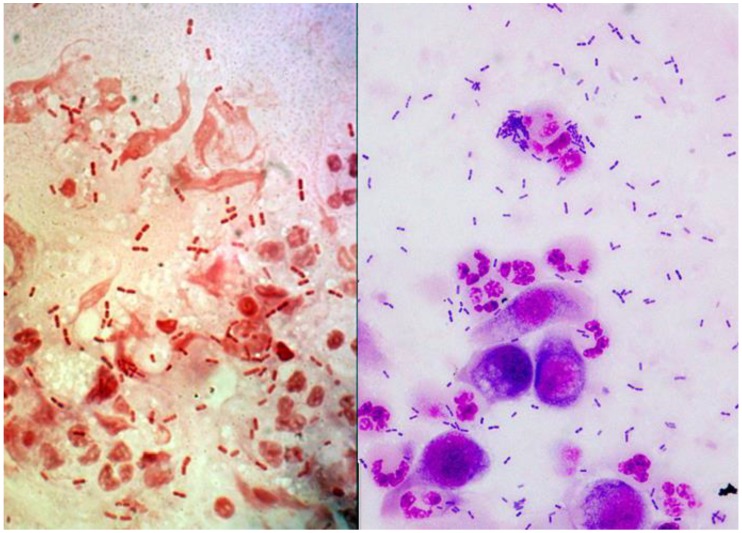
The presence of *Moraxella* diplobacilli from corneal scrapings using Gram stain (left picture) and Giemsa (right picture). The pictures were photographed under (100× oil immersion).

**Figure 2 microorganisms-07-00163-f002:**
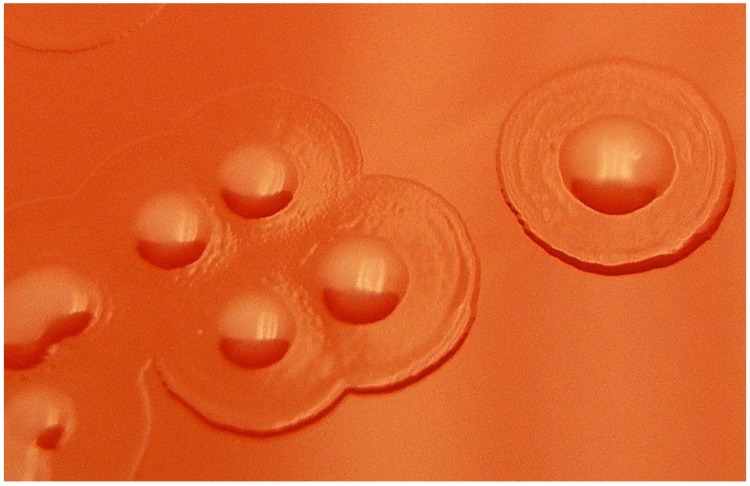
Fried egg appearance of *Moraxella* growing on trypticase soy agar supplemented with 5% sheep blood. The colonies were magnified by 20×.

**Figure 3 microorganisms-07-00163-f003:**
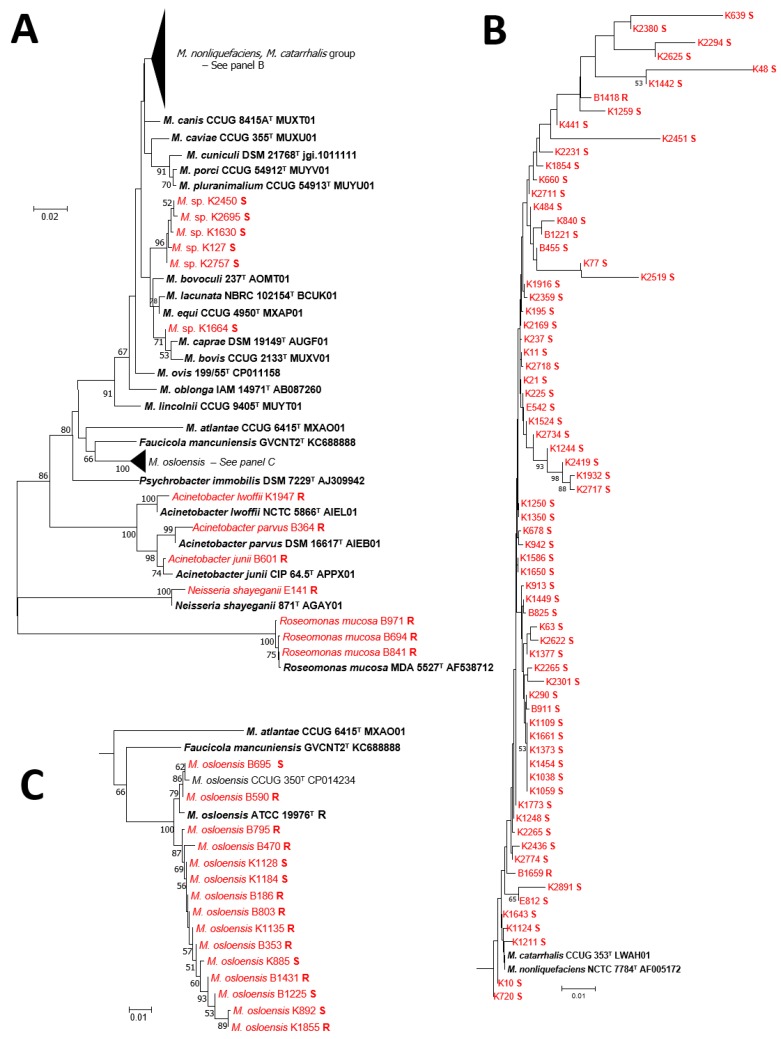
A diagram of the neighbor joining tree of 16S rRNA sequences for the *Moraxella* isolates in the study. Panel (**A**) details the neighbor joining tree (25) constructed with all sequences from this study and relevant type strains. Clades corresponding to *M. nonliquefaciens/M.catarrhalis* and *M. osloensis* are condensed. Panel (**B**) depicts clades corresponding to *M. nonliquefaciens/M.catarrhalis.* Panel (**C**) corresponds to *M. osloensis.* Red indicates strains from our clinical collection and black indicates select type strains. The vancomycin susceptibility status is indicated as S for susceptible and R for resistant.

**Table 1 microorganisms-07-00163-t001:** Identification of *Moraxella* species from keratitis, conjunctivitis, and endophthalmitis using DNA sequencing, Biolog, and MALDI-TOF MS.

Isolate	Van S/R	Identification Based on: (DNA Sequencing with Vancomycin Susceptibility)	Biolog ID	MALDI-TOF MS
**ATCC Controls**				
*Moraxella bovis* 10900	S	*M. bovis*	*M. bovis*	*M. bovis*
*M. caviae* 14659	R	*M. caviae*	*M. caviae*	No ID
*M. cuniculi* 14688	R	*M. cuniculi*	*M. cuniculi*	*M. catarrhalis*
*M. nonliquefaciens* 19975	S	*M. nonliquefaciens*	No ID	*M. nonliquefaciens*
*M. osloensis* 19976	R	*M. osloensis*	*M. osloensis*	*M. osloensis*
*M. atlantae* 29525	I	*M. atlantae*	No ID	*M. atlantae*
*M. lincolnii* 51388	Q	*M. lincolnii*	No ID	*M. lincolnii*
*M. lacunata* 17967	Q	*M. lacunata*	No ID	*M. lacunata*
*M. catarrhalis* 24250	R	*M. catarrhalis*	*M. catarrhalis*	*M. catarrhalis*
**Keratitis Isolates**				
1. K10	S	*M. nonliquefaciens*	*M. nonliquefaciens*	*M. nonliquefaciens*
2. K11	S	*M. nonliquefaciens*	*M. nonliquefaciens*	*M. nonliquefaciens*
3. K21	S	*M. nonliquefaciens*	*M. nonliquefaciens*	*M. nonliquefaciens*
4. K48	S	*M. nonliquefaciens*	*M. nonliquefaciens*	*M. nonliquefaciens*
5. K63	S	*M. nonliquefaciens*	*M. equi*	*M. nonliquefaciens*
6. K77	S	*M. nonliquefaciens*	*M. nonliquefaciens*	*M. nonliquefaciens*
7. K127	S	*M. equi, M. lacunata, M. bovoculi*	*M. nonliquefaciens*	*M. lacunata, M. bovis*
8. K195	S	*M. nonliquefaciens*	*M. nonliquefaciens*	*M. nonliquefaciens*
9. K225	S	*M. nonliquefaciens*	*M. nonliquefaciens*	*M. nonliquefaciens*
10. K237	S	*M. nonliquefaciens*	*M. nonliquefaciens*	*M. nonliquefaciens*
11. K290	S	*M. nonliquefaciens*	*M. nonliquefaciens*	*M. nonliquefaciens*
12. K441	S	*M. nonliquefaciens*	*M. nonliquefaciens*	*M. nonliquefaciens*
13. K484	S	*M. nonliquefaciens*	*M. nonliquefaciens*	*M. nonliquefaciens*
14. K639	S	*M. nonliquefaciens*	*M. nonliquefaciens*	*M. nonliquefaciens*
**Keratitis Isolate**				
15. K660	S	*M. nonliquefaciens*	*M. nonliquefaciens*	*M. nonliquefaciens*
16. K678	S	*M. nonliquefaciens*	*M. nonliquefaciens*	*M. nonliquefaciens*
17. K720	S	*M. nonliquefaciens*	*M. nonliquefaciens*	*M. nonliquefaciens*
18. K840	S	*M. nonliquefaciens*	*M. equi*	*M. nonliquefaciens*
19. K885	S	*M. osloensis*	*M. canis*	*M. osloensis*
20. K892	S	*M. osloensis*	*M. osloensis*	*M. osloensis*
21. K913	S	*M. nonliquefaciens*	*M. nonliquefaciens*	*M. nonliquefaciens*
22. K942	S	*M. nonliquefaciens*	*M. nonliquefaciens*	*M. nonliquefaciens*
23. K1038	S	*M. nonliquefaciens*	*M. equi*	*M. nonliquefaciens*
24. K1059	S	*M. nonliquefaciens*	*M. bovis*	*M. nonliquefaciens*
25. K1109	S	*M. nonliquefaciens*	*M. nonliquefaciens*	*M. nonliquefaciens*
26. K1124	S	*M. nonliquefaciens*	*M. nonliquefaciens*	*M. nonliquefaciens*
27. K1128	S	*M. osloensis*	*M. osloensis*	*M. osloensis*
28. K1135	R	*M. osloensis*	*M. osloensis*	*M. osloensis*
29. K1184	S	*M. osloensis*	*M. nonliquefaciens*	*M. nonliquefaciens*
30. K1211	S	*M. nonliquefaciens*	*M. nonliquefaciens*	*M. nonliquefaciens*
31. K1219	S	*M. nonliquefaciens*	No ID	No ID
32. K1244	S	*M. nonliquefaciens*	*M. nonliquefaciens*	*M. nonliquefaciens*
33. K1248	S	*M. nonliquefaciens*	*M. nonliquefaciens*	*M. nonliquefaciens*
34. K1250	S	*M. nonliquefaciens*	*M. nonliquefaciens*	*M. nonliquefaciens*
35. K1259	S	*M. nonliquefaciens*	*M. nonliquefaciens*	*M. nonliquefaciens*
36. K1350	S	*M. nonliquefaciens*	*M. nonliquefaciens*	*M. nonliquefaciens*
37. K1361B	R	*Acinetobacter lwoffii*	*M. osloensis*	*Acinetobacter lwoffii*
38. K1373	S	*M. nonliquefaciens*	*M. catarrhalis*	*M. nonliquefaciens*
39. K1377	S	*M. nonliquefaciens*	No ID	No ID
40. K1442	S	*M. nonliquefaciens*	*M. nonliquefaciens*	*M. nonliquefaciens*
41. K1449	S	*M. nonliquefaciens*	No ID	No ID
42. K1454	S	*M. nonliquefaciens*	*M. nonliquefaciens*	*M. nonliquefaciens*
43. K1524	S	*M. nonliquefaciens*	*M. equi*	*M. nonliquefaciens*
44. K1586	S	*M. nonliquefaciens*	*M. equi*	*M. nonliquefaciens*
45. K1630	S	*M. equi, M. bovoculi, M. lacunata*	*M. equi*	*M. lacunata*
46. K1643	S	*M. nonliquefaciens*	*M. nonliquefaciens*	*M. nonliquefaciens*
47. K1650	S	*M. nonliquefaciens*	*M. nonliquefaciens*	*M. nonliquefaciens*
48. K1661	S	*M. nonliquefaciens*	*M. nonliquefaciens*	*M. nonliquefaciens*
49. K1664	S	*M. equi, M. bovoculi, M. lacunata*	No ID	No ID
50. K1773	S	*M. nonliquefaciens*	*M. nonliquefaciens*	*M. nonliquefaciens*
51. K1784	S	*M. equi, M. bovoculi, M. lacunata*	*M. ovis*	No ID
52. K1661	S	*M. nonliquefaciens*	*M. nonliquefaciens*	*M. nonliquefaciens*
53. K1854	S	*M. nonliquefaciens*	*M. nonliquefaciens*	*M. nonliquefaciens*
54. K1855	R	*M. osloensis*	*M. osloensis*	*M. osloensis*
55. K1916	S	*M. nonliquefaciens*	*M. nonliquefaciens*	*M. nonliquefaciens*
56. K1932	S	*M. nonliquefaciens*	*M. caprae*	
57. K1947	R	*Acinetobacter lwoffii*	*Acinetobacter lwoffi*	*Acinetobacter lwoffii*
58. K2169	S	*M. nonliquefaciens*	*M. nonliquefaciens*	*M. nonliquefaciens*
59. K2231	S	*M. nonliquefaciens*	*M. nonliquefaciens*	*M. nonliquefaciens*
60. K2265	S	*M. nonliquefaciens*	*M. nonliquefaciens*	*M. nonliquefaciens*
61. K2275	S	*M. nonliquefaciens*	*M. nonliquefaciens*	*M. nonliquefaciens*
62. K2294	S	*M. nonliquefaciens*	*M. nonliquefaciens*	*M. nonliquefaciens*
63. K2301	S	*M. nonliquefaciens*	*M. nonliquefaciens*	*M. nonliquefaciens*
64. K2359	S	*M. nonliquefaciens*	*M. nonliquefaciens*	*M. nonliquefaciens*
65. K2380	S	*M. nonliquefaciens*	*M. nonliquefaciens*	*M. nonliquefaciens*
66. K2419	S	*M. nonliquefaciens*	*M. nonliquefaciens*	*M. nonliquefaciens*
67. K2436	S	*M. nonliquefaciens*	*M. equi*	
68. K2450	S	*M. equi, M. bovoculi, M. lacunata*	*M. nonliquefaciens*	*M. lacunata*
69. K2451	S	*M. nonliquefaciens*	*M. nonliquefaciens*	*M. nonliquefaciens*
70. K2519	S	*M. nonliquefaciens*	*M. nonliquefaciens*	*M. nonliquefaciens*
71. K2565	S	*M. nonliquefaciens*	*M. nonliquefaciens*	*M. nonliquefaciens*
72. K2622	S	*M. nonliquefaciens*	*M. nonliquefaciens*	*M. nonliquefaciens*
73. K2625	S	*M. nonliquefaciens*	*M. nonliquefaciens*	*M. nonliquefaciens*
74. K2695	S	*M. equi, M. bovoculi, M. lacunata*	*M. nonliquefaciens*	*M. lacunata*
75. K2711	S	*M. nonliquefaciens*	*M. nonliquefaciens*	*M. nonliquefaciens*
76. K2717	S	*M. nonliquefaciens*	*M. nonliquefaciens*	*M. nonliquefaciens*
77. K2718	S	*M. nonliquefaciens*	*M. bovis*	*M. bovis*
78. K2734	S	*M. nonliquefaciens*	*M. nonliquefaciens*	*M. nonliquefaciens*
79. K2757	S	*M. equi, M. bovoculi, M. lacunata*	No ID	*M. lacunata*
80. K2774	S	*M. nonliquefaciens*	*M. nonliquefaciens*	*M. nonliquefaciens*
81. K2880	S	*M. nonliquefaciens*	*M. nonliquefaciens*	*M. nonliquefaciens*
82. K2891	S	*M. nonliquefaciens*	*M. nonliquefaciens*	*M. nonliquefaciens*
**Conjunctivitis Isolates**				
1. B186	R	*M. osloensis*	*M. osloensis*	*M. osloensis*
2. B353	R	*M. osloensis*	*M. osloensis*	*M. osloensis*
3. B364	R	*Acinetobacter parvus*	No ID	*Acinetobacter parvus*
4. B455	S	*M. nonliquefaciens*	*M. nonliquefaciens*	*M. nonliquefaciens*
5. B470	R	*M. osloensis*	*M. osloensis*	*M. osloensis*
6. B590	R	*M. osloensis*	*M. osloensis*	No ID
7. B601	R	*Acinetobacter junii*	*Acinetobacter junii*	*Acinetobacter junii*
8. B662	S	*M. nonliquefaciens*	*M. nonliquefaciens*	*M. nonliquefaciens*
9. B694	R	*Roseomonas mucosa*	No ID	*Roseomonas mucosa*
10. B695	S	*M. osloensis*	*M. osloensis*	*M. osloensis*
11. B795	R	*M. osloensis*	*M. canis*	*M. osloensis*
12. B803	R	*M. osloensis*	No ID	*M. osloensis*
13. B825	S	*M. nonliquefaciens*	No ID	No ID
14. B841	R	*Roseomonas mucosa*	No ID	*Roseomonas mucosa*
15. B911	S	*M. nonliquefaciens*	*M. nonliquefaciens*	*M. nonliquefaciens*
16. B971	R	*Roseomonas mucosa*	No ID	*Roseomonas mucosa*
17. B1221	S	*M. nonliquefaciens*	*M. nonliquefaciens*	*M. nonliquefaciens*
18. B1225	S	*M. osloensis*	*M. bovis*	*M. osloensis*
19. B1418	R	*M. nonliquefaciens*	*M. equi*	*M. nonliquefaciens*
20. B1431	R	*M. osloensis*	*M. osloensis*	*M. osloensis*
21. B1659	R	*M. nonliquefaciens*	*M. catarrhalis*	*M. catarrhalis*
**Endophthalmitis Isolates**				
1. E141	R	*Neisseria shayeganii*	No ID	No ID
2. E542	S	*M. nonliquefaciens*	*M. nonliquefaciens*	*M. nonliquefaciens*
3. E614	S	*M. nonliquefaciens*	*M. bovis*	*M. nonliquefaciens*
4. E812	S	*M. nonliquefaciens*	*M. nonliquefaciens*	*M. nonliquefaciens*

Vancomycin (Van) susceptibility was determined using the standard Kirby–Bauer disk diffusion method. Zones greater than 18 mm were interpreted as susceptible (S) and less were resistant (R). *M. atlantae* had a zone of 16 mm. “I” is intermediate and “Q” is questionable. “No ID” indicated that no identification was made. Biolog is an identification system that uses 94 biochemicals to phenotypically fingerprint bacterial isolates. MALDI-TOF MS is matrix-assisted laser desorption ionization time-of-flight mass spectrometry.

**Table 2 microorganisms-07-00163-t002:** The Identification of *Moraxella* to species as isolated from keratitis, conjunctivitis, and endophthalmitis (1993–2016).

**Keratitis (*n* = 82)**	**Incidence (percent)**
*Moraxella nonliquefaciens*	66 (80.5%)
*Moraxella lacunata*	7 (9.0%)
*Moraxella osloensis*	5 (6.0%)
*Acinetobacter lwoffii*	2 (2.5%)
*Moraxella bovis/nonliquefaciens*	1 (1.0%)
*Moraxella osloensis/nonliquefaciens*	1 (1.0%)
**Conjunctivitis (*n* = 21)**	
*Moraxella osloensis*	9 (43.0%)
*Moraxella nonliquefaciens*	6 (29.0%)
*Roseomonas mucosa*	3 (14.0%)
*Acinetobacter* (*parvus*, *junii*)	2 (9.5%)
*Moraxella catarrhalis/nonliquefaciens*	1 (4.5%)
**Endophthalmitis (*n* = 4)**	
*Moraxella nonliquefaciens*	3 (75%)
*Neisseria shayeganii*	1 (25%)
All Infections	
*Moraxella nonliquefaciens*	70% (75 of 107)
*Moraxella osloensis*	13% (14 of 107)

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
