# Peer review of "Moraxella nonliquefaciens and M. osloensis Are Important Moraxella Species That Cause Ocular Infections"

_microorganisms, 2019, doi:10.3390/microorganisms7060163_

Round 1

Reviewer 1 Report

This is an interesting paper comparing microbiological methods for detection and identification of Moraxella species recovered from ocular sources. There are some concerns..

First , Identification of Moraxella species from ocular samples based on morphology alone can be challenging and sometimes inaccurate. Members of this group can be diplobaccill, diplococci and or coccobacillary on gram stain. Including oxidase, growth on chocolate and or Macconkey agars can help separate the group from Acinetobacter and other similar organisms. 

This  is a retrospective study, so Table 2 should indicate prevalence rather than incidence.  

It is not clear how the authors determined that 70% (75/107) of the isolates were M. nonliquefaciens  and 13% (14 of 107) were M. osloensis.  DNA sequencing results were 72% (77/107) and 14% (15/107)  respectively, while MALDI-TOF was 67. 3% (72/1107) and 12.1% (13/107) respectively. Authors should identify how they determined the composite numbers.

 Although overall agreement between DNA sequencing and MALDI-TOF was 91%, unequivocal identification or confirmation of  the isolates  as Moraxella species were only 86% (92/107) and 

79% (85/107) respectively. 

It is a bit surprising that the authors did not recover more M. lacunata  from these specimens and or accept this identification from proteomics (M-TOF).  This lack of identification may have been due to the probed and available databases (nucleotide vs protein). Authors should include this information in their discussion. 

Other rapid, commercial systems are available for identification of Moraxella species from clinical samples including Remel kits and Vitek identification cards. Discussion on the possible use of these or the reported results using these could have been useful.

 It is clear from these results, that identification of Moraxella organisms to the species level remain a challenge. 

Author Response

I am sending a revision table that includes comments to both reviewers and editor

Reviewer 2 Report

Moraxella nonliquefaciens as well as Morxella osloensis are well known as skin commensals.

For this reason the authors have to prove that Moraxella isolates stored for up to 25 years in a bacteria collection are mainly real pathogens and not only bystanders.

This seems to be impossible without inclusion of clinical data of patients and bacteriological reevaluation of the materials taken from the patients since some pathogens might be suppressed with techniques used 25 years ago.

Additionally, any whole genome technique should be employed for exclusion clonal strain clusters of M. nonliquefaciens or M. osloensis strains in the collection investigated.

Line 133: What is ALDI-TOF MS?

The numbers of Table 2 are not quite correct.

Author Response

I have uploaded a file that includes the responses to both reviewers and the editor

Round 2

Reviewer 2 Report

The authors did not solved the question if the isolated Moraxella species which are known as skin commensals are really eye pathogens.

According the Koch-Henley postulates, they should show by FISH or related techniques in original swabs or clinical samples that the amount and distribution of the Moraxella nonliquefaciens as well as Moraxella osloensis justify them as eye pathogens.

Author Response

As indicated by my previous notes to editor: 1) We fixed all typos, miss spellings, and grammar mistakes as requested. These were yellow highlighted in the revised text. 2) For lines 61-62, we added the sentence, " The cases of keratitis, endophthalmitis, and conjunctivitis were treated as infections by Moraxella". This should satisfy the reviewers question of Moraxella as a pathogen and not a skin commensuals. Moraxella was isolated from the eye. 3)Table 1 was corrected in regards to periods after the M and the genus and species were italicized. Table ! was created as a separate file. 4) The website was added on line 290. 5) Table 2 was corrected in regards to periods after the M and the genus and species were italized. Table 2 was created as a separate file.
